DOI: 10.1038/s41467-018-07352-1　　**OPEN**

# Vortex rectenna powered by environmental fluctuations

J. Lustikova[1], Y. Shiomi[1,7], N. Yokoi [1], N. Kabeya [2,3], N. Kimura[2,3], K. Ienaga[4], S. Kaneko[4], S. Okuma[4], S. Takahashi[1] & E. Saitoh[1,5,6,8]

A rectenna, standing for a rectifying antenna, is an apparatus which generates d.c. electricity from electric fluctuations. It is expected to realize wireless power transmission as well as energy harvesting from environmental radio waves. To realize such rectification, devices that are made up of internal atomic asymmetry such as an asymmetric junction have been necessary so far. Here we report a material that spontaneously generates electricity by rectifying environmental fluctuations without using atomic asymmetry. The sample is a common superconductor without lowered crystalline symmetry, but, just by putting it in an asymmetric magnetic environment, it turns into a rectifier and starts generating electricity. Superconducting vortex strings only annihilate and nucleate at surfaces, and this allows the bulk electrons to feel surface fluctuations in an asymmetric environment: a vortex rectenna. The rectification and generation can be switched on and off with only a slight change in temperature or external magnetic fields.

[1] Institute for Materials Research, Tohoku University, Sendai 980-8577, Japan. [2] Graduate School of Science, Tohoku University, Sendai, Miyagi 980-8578, Japan. [3] Center for Low Temperature Science, Tohoku University, Sendai 980-8578, Japan. [4] Department of Physics, Tokyo Institute of Technology, Meguro, Tokyo 152-8551, Japan. [5] Advanced Institute for Materials Research, Tohoku University, Sendai 980-8577, Japan. [6] Advanced Science Research Center, Japan Atomic Energy Agency, Tokai 319-1195, Japan. [7] Present address: Department of Applied Physics and Quantum-Phase Electronics Center (QPEC), The University of Tokyo, Hongo, Tokyo 113-8656, Japan. [8] Present address: Department of Applied Physics, The University of Tokyo, Hongo, Tokyo 113-8656, Japan. Correspondence and requests for materials should be addressed to E.S. (email: eizi@ap.t.u-tokyo.ac.jp)

A typical rectification device is a diode, which conducts electricity primarily in one direction[1]. It is typically composed of different types of semiconductors connected to form an atomic junction called a p–n junction, exhibiting low resistance in one direction while high in the other. This asymmetry can convert a.c. current fluctuations passing through the junction into a d.c. current.

Not only an atomic junction, theoretically, but a conductor itself may exhibit such a diode property even without junctions. The required condition is as follows: first, an atomic crystal structure should be asymmetric, that is, its unit cell should be composed of a highly asymmetric arrangement of atoms or ions[2]. Second, the sample should exhibit magnetization. In fact, nonlinear transport was observed in some magnetized materials in which crystalline inversion symmetry is broken[3–8].

In this way, some atomic-scale asymmetry looks indispensable for rectification so that electrons can feel the asymmetry: but is it always true? Cannot rectification be realized just by putting a symmetric sample in an asymmetric environment? Intuitively, it seems impossible in conventional conductors, since the bulk of electrons cannot feel the outside of the sample due to their very short coherence length. However, it is notable that a superconductor maintains macroscopic quantum order, which may allow electrons to feel fluctuations far away.

In three-dimensional superconducting order, local excitations are carried by topological objects called vortex strings: a string-shaped normal core around which a supercurrent whirls[9]. Vortex strings are topologically protected; it is not possible to annihilate a vortex string by a continuous transformation of the superconducting order parameter. Vortex strings are then never created or annihilated inside a sample[10]. Surfaces of a superconductor are the only locations where a vortex can nucleate and be annihilated due to local electromagnetic fluctuations. Even so, as far as the fluctuation is uniform, such nucleation and annihilation should compensate among the surfaces, and no net flow of vortex strings appears. On the other hand, if asymmetry of the electromagnetic environment around the surface of superconductor exists, it may break the compensation and give rise to a finite net flow of vortex strings (Fig. 1). Finally, the vortex string flow causes the appearance of an electric field in the direction perpendicular to the flow direction[11]. Therefore, owing to the

conservation of vortex flow in bulk, surface electromagnetic fluctuations are expected to be transcribed into a vortex-string current, and consequently into a voltage when the fluctuation property is different among the surfaces, realizing electric power generation from nonequilibrium environment fluctuations[12–14].

In this work, we do not use vortex ratchet patterns[15–18], but a simple boundary condition for vortex nucleation is used as proposed theoretically by Vodolazov and Peeters[19]. Although such rectification was studied by edge roughness in superconducting strips[20,21], we are not aware of any previous works that report power generation from environmental fluctuations.

## Results

**Sample and measurement setup**. Figure 2a is a schematic illustration of the sample system used in the present study. The sample is a thick (350 nm) film of amorphous MoGe (Fig. 2b); a typical type-II superconductor that exhibits a sharp transition to a clean vortex liquid phase, where vortex strings can flow freely[22–24]. A MoGe film was sputtered on an insulator magnet $Y_3Fe_5O_{12}$ (YIG). The bottom surface of the MoGe film is exposed close to YIG, while the top surface to a vacuum. YIG exhibits much greater magnetic susceptibility $\chi$ than a vacuum, giving rise to a difference in magnetic components of the fluctuation amplitude between the bottom and the top surfaces of the MoGe. In the vortex liquid phase, the difference in the fluctuations is transcribed into a net vortex-string flow perpendicular to the film plane. This vortex flow can be detected by measuring the voltage difference between ends of the MoGe film in the transverse direction[9].

Rectification of current can be evaluated by measuring nonlinearity of resistivity. In a material rectifying current, the resistivity $\rho$ should exhibit nonreciprocity, that is, $\rho(I) \neq \rho(-I)$. When the current $I$ is small enough, the resistivity can be expanded as $\rho(I) = \rho_0 + \Delta_{2f}I + \cdots$, where $\Delta_{2f}$ represents the asymmetry of $\rho$ with respect to $I$, and $\rho(I) \neq \rho(-I)$ is thus transcribed into $\Delta_{2f} \neq 0$. Hence, $\Delta_{2f}$ can be used as a measure of current rectification. $\Delta_{2f}$ can be precisely measured by a 2f-lock-in method for a voltage generated in the sample, $V = \rho(I)I\ell/S = \left(\rho_0 I + \Delta_{2f}I^2 + \cdots\right)\ell/S$, by applying a small a.c. current, $I$, with the frequency $f$, and detecting a generated voltage with the doubled frequency $2f$ between electrodes separated by the distance $\ell$ in a specimen with the cross-section area $S$.

**Phase diagram of MoGe**. Figure 2c is a phase diagram of superconductivity in the present MoGe|YIG sample, with vortex melting field $B_m$ and upper critical field $B_{c2}$ defined as in ref. [25]. Here, the magnetic field is almost parallel to the sample surface. Below $T_c = 6.9$ K, a superconducting phase appears exhibiting zero resistivity. By applying a magnetic field, $B$, a vortex liquid phase (yellow region in Fig. 2c) appears when $B_m \leq B \leq B_{c2}$. In the liquid phase, vortex strings can move almost freely in a superconductor, and finite resistance appears in spite of the superconductivity. In Fig. 2d, we exemplify resistance of the MoGe layer as a function of $B$ at $T = 4$ K. At $B = 0$, resistivity $\rho_0$ is zero. By applying a magnetic field, finite resistivity appears around $B \sim 7$ T, signaling a transition into a vortex liquid phase, and, by further increasing $B$, the MoGe layer turns into a normal metal phase around $B_{c2} \sim 8$ T.

**Observation of nonreciprocity in vortex liquid phase**. In the vortex liquid phase, we found clear current-rectification. Figure 2e shows $\Delta_{2f}$ observed for the MoGe layer in the MoGe|YIG. Importantly, a clear signal of $\Delta_{2f}$ appears at $6.5$ T $< B < 8$ T, forming a sharp peak. On the other hand, $\Delta_{2f}$ is almost zero in the

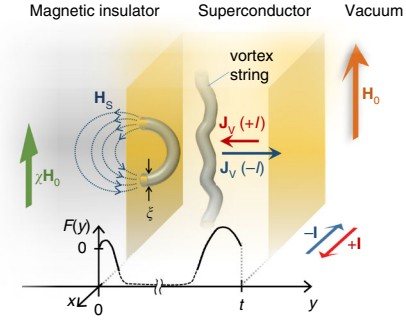

**Fig. 1** Nonreciprocity in a system with topological objects in an asymmetric external environment. A superconductor with vortex strings close to a magnetic insulator with susceptibility $\chi$ is placed in an external magnetic field $\mathbf{H}_0$. Vortex strings are depicted as normal cores in the superconducting condensate, with a characteristic diameter proportional to the superconducting coherence length $\xi$. Stray magnetic fields $\mathbf{H}_S$ emanate from vortex strings nucleating at the superconductor surfaces. The difference in the nucleation energy $F$ for vortex strings at the interface with the magnetic insulator ($y = 0$) and the vacuum ($y = t$) leads to a nonreciprocity in the vortex flow $\mathbf{J}_V$ when driving the vortex strings with an electric current $\pm \mathbf{I}$

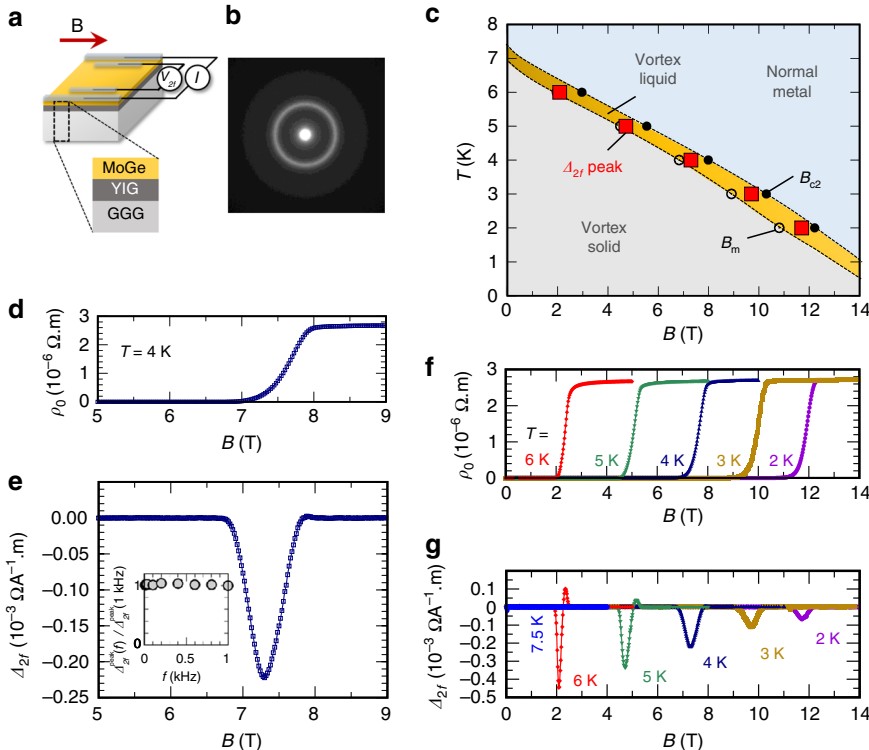

**Fig. 2** Observation of nonreciprocal transport in MoGe|YIG. **a** A schematic illustration of the sample structure and measurement setup. The MoGe| $Y_3Fe_5O_{12}$ (YIG) sample on a $Gd_3Ga_5O_{12}$ (GGG) substrate is placed in a magnetic field almost parallel to the sample surface. An a.c. input current $I$ with a frequency $f$ is applied to the sample, and $V_{2f}$, the a.c. voltage signal with frequency $2f$, is measured. **b** Electron diffraction pattern from the MoGe layer. **c** The $B$–$T$ phase diagram of the MoGe and appearance of non-zero nonreciprocal resistivity denoted as $\Delta_{2f} \neq 0$ peak. The vortex-solid melting field $B_m$ and the upper critical field $B_{c2}$ were determined by the conditions $\rho(B_m) = 10^{-3}\rho_N$ and $\rho(B_{c2}) = 0.95\rho_N$, respectively, where $\rho_N$ is the resistivity in the normal state. Dotted lines are guides for eyes. **d** Magnetic field dependence of resistivity $\rho_0$ at $T = 4$ K. **e** Magnetic field dependence of nonreciprocal resistivity coefficient $\Delta_{2f}$ ($I_0 = 40\,\mu$A, $f = 13.7$ Hz). Inset: Frequency dependence of the peak value of $\Delta_{2f}$ divided by the peak value at $f = 1$ kHz, $\Delta_{2f}^{peak}/\Delta_{2f}^{peak}$(1 kHz). Magnetic field dependence of **f** $\rho_0$, and **g** $\Delta_{2f}$ at selected temperatures

superconducting state ($B < 6.5$ T) and the normal state ($B > 8$ T). The peak value of $\Delta_{2f}$ does not change with a change in the frequency of $I$ (inset to Fig. 2e). Figure 2c shows the position of the $\Delta_{2f}$ peaks on the $T$–$B$ phase diagram of the MoGe. $\Delta_{2f}$ appears just in the vortex liquid phase where the mobility of vortex strings is maximal. The appearance of $\Delta_{2f}$ in the vortex liquid phase indicates the nonreciprocity, $\rho(I) \neq \rho(-I)$, in spite of the isotropic atomic structure in MoGe. Such nonreciprocity can, in principle, give rise to rectification of nonequilibrium environmental electromagnetic fluctuation (e.g., colored current noise)[12–14], which has been difficult to observe in electron systems. However, thanks to the extremely sharp transition into the nonreciprocal vortex liquid phase in the present system, we successfully demonstrate power generation by rectifying environmental fluctuations; a small change in the magnetic field or temperature facilitates very steep switching of $\Delta_{2f}$.

**Measurement of d.c. power generation**. We measured spontaneous electric power generation in the MoGe|YIG by putting the sample into one of the most commonly used measurement systems: a Physical Property Measurement System (PPMS, Quantum Design Inc.). Figure 3 shows the d.c. voltage spontaneously generated in the MoGe film of the MoGe|YIG sample without applying any external currents. Surprisingly, clear voltage peaks appear at fields corresponding to the peaks of $\Delta_{2f}$ in the vortex liquid phase (the yellow area in Fig. 3a).

Magnetic field dependence of the d.c. voltages at selected temperatures is shown in Fig. 3b. In the metallic phase at $T = 7.5$

K $> T_c$, no $V_{dc}$ was observed. Below $T_c$, the polarity of $V_{dc}$ is reversed by reversing the direction of the magnetic field. This is consistent with the field dependence of flux flow resistivity, implying that the voltage is caused by vortex motion. Interestingly, we found that a constant value of $V_{dc}$ persists over the course of hours or even days when $B$ and $T$ are fixed in the vortex liquid phase (see Supplementary Fig. 5 and Supplementary Note 2). By switching the superconducting magnet to the persistent mode where it is disconnected from the power source, $V_{dc}$ decreases a little but remains finite (see Supplementary Fig. 6 and Supplementary Note 2). We confirmed that the generated voltage can perform work by measuring the power on an external resistive load connected to the sample (Fig. 3c).

Residual temperature gradients in the sample are irrelevant to the observed voltage. We show that external temperature gradients do not affect the d.c. voltage signal of interest (Fig. 3d–f). The ambient temperature of the film was kept at $T = 4.02$ K and the temperature difference $\Delta T$ between the top of the MoGe and the bottom of the substrate was varied. For $\Delta T = +0.21$ K, the voltage spectrum consists of a broad shoulder of positive polarity ($8$ T $\leq B \leq 14$ T) and a sharp voltage peak of negative polarity at $\sim$8 T (for $B > 0$, Fig. 3d). When the temperature gradient is turned off (Fig. 3e), only the broad shoulder part disappears, and the single sharp voltage peak at $\sim$8 T still subsists. For a negative temperature gradient $\Delta T = -0.21$ K, the voltage spectrum consists again of a broad shoulder at the higher magnetic field and a sharp voltage peak at $\sim$8 T (Fig. 3f). The polarity of the broad shoulder voltage at the higher magnetic fields is reversed by reversing the polarity of $\Delta T$, showing that it is

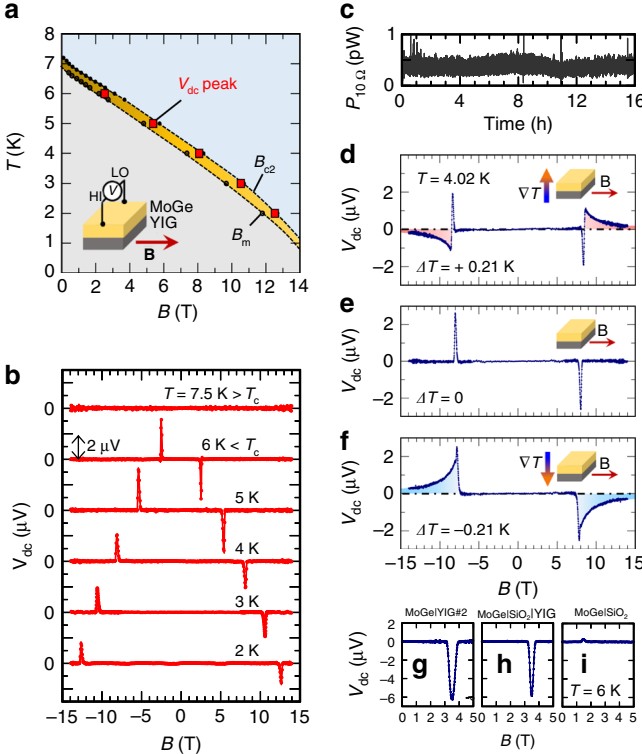

**Fig. 3** D.c. voltage generation observed in MoGe|YIG. **a** D.c. voltage $V_{dc}$ peak in the $B$–$T$ phase diagram of MoGe. The inset shows the measurement setup. The direction of the magnetic field is almost parallel to the sample surface. **b** Magnetic field dependence of the d.c. voltage generated in the MoGe layer at selected temperatures. The length of the scale bar is 2 μV. **c** Temporal evolution of the generated power $P_{10\Omega}$ in a constant magnetic field ($T = 6$ K, $B = 2.5$ T, with the magnet in persistent mode). $P_{10\Omega}$ was measured on a 10 Ω resistor connected parallel to the sample. **d**–**f** Magnetic field dependence of d.c. voltages in a MoGe|YIG sample for positive, zero, and negative temperature difference, respectively, between the top and the bottom of the sample at $T = 4$ K. D.c. voltage generated in **g** MoGe|YIG sample #2, **h** MoGe|SiO$_2$(10 nm)|YIG, and **i** MoGe|SiO$_2$ at $T = 6$ K

a contribution of the vortex Nernst effect: voltage due to the motion of vortex strings induced by the thermal gradient[11,26]. By contrast, reversing the polarity of $\Delta T$ does not affect the sharp peak at ~8 T proving that it is not related to temperature gradients or vortex Nernst effects, but it is consistent with the above rectification mechanism.

**Influence of substrate choice on nonreciprocity in MoGe.** Importantly, $\Delta_{2f}$ and the spontaneous voltage generation do not change when an insulating SiO$_2$ layer is inserted between the YIG and the MoGe layers to form MoGe|SiO$_2$(10 nm)|YIG (Fig. 3g, h, see also Supplementary Fig. 8 and Supplementary Note 2). The amplitude of $V_{dc}$ turned out to be comparable to that observed in the MoGe|YIG bilayers. This means that the observed spontaneous voltage generation and nonreciprocal transport appear just when MoGe is put close to a magnet YIG; direct contact is not necessary. Therefore, the relevant mechanism should be magnetic coupling between YIG and MoGe, which is not affected by a thin insulating barrier. When the ferrimagnetic YIG is removed and a MoGe film is deposited directly on an oxidized Si substrate which exhibits diamagnetism (MoGe|SiO$_2$ in Fig. 3i), the voltage generation almost disappears, and a very weak signal of opposite sign is observed. This implies that the diamagnetism of SiO$_2$ induces a similar effect but with opposite polarity. The same result has been obtained on diamagnetic Y$_3$Al$_5$O$_{12}$ substrates, which have a morphology similar to YIG (see Supplementary Fig. 9 and Supplementary Note 2).

**Theoretical model for nonreciprocity in vortex liquid phase.** We calculated vortex flow in the vortex liquid phase based on the

Ginzburg–Landau (GL) effective theory of superconductivity. For simplicity, we assume that the amount of the vortex flow induced by current bias in the vortex liquid phase is dominated by nucleation process[20,21,24] of the vortex strings at the interfaces. According to the GL effective description, the solitonic vortex strings are nucleated dominantly as a semicircle form at the interfaces[27,28]. The nucleation rate $P$ of a vortex is given by the energy barrier $\Delta F$, $P \sim \Omega e^{-\Delta F/k_B T}$, where $\Omega$ is a frequency prefactor, $k_B$ is the Boltzmann constant, and $T$ represents temperature[29,30]. A nucleated vortex semicircle at the MoGe/YIG interface produces the stray magnetic field $\mathbf{H}_S$ in the YIG substrate to close the magnetic flux of the vortex (Fig. 1), and the magnetic energy in YIG gives an extra contribution to the vortex energy via the large transverse susceptibility $\chi_\perp$ in YIG. The extra contribution $F_{mag}$ reduces the vortex energy, and induces a difference in the nucleation energy at the interfaces, $\Delta F_{YIG} < \Delta F_{vac}$. The subscripts YIG and vac mean the values at the interfaces to the YIG layer and to the vacuum, respectively. As a result, this gives a difference in the nucleation rates between the MoGe/YIG and MoGe/vacuum interfaces, and leads to a difference proportional to $\chi_\perp - \chi_{vac}$ in the vortex flow between the directions away from and toward the YIG substrate under the external currents $-I$ and $+I$, respectively. Here, the current $I$ exerts the Lorentz force on the vortex, and we used the conservation of the vortex topological number in the bulk. The vortex flow should be zero in the absence of $I$, which can be realized by the accumulation[31] of the vortex strings. In the vortex liquid phase, the mobility of the vortex strings is maximal, and the nonreciprocal response to external currents is expected to be the largest. This result is consistent with the observed nonreciprocity of the vortex flow

resistivity in the vortex liquid phase, $\rho(I) \neq \rho(-I)$, and with the observed change in the d.c. voltage signal between the YIG and the $SiO_2$ substrates (see Supplementary Note 1).

## Discussion

Nonreciprocal resistivity in the vortex liquid phase of the amorphous superconductor MoGe attached to a magnetic insulator has been observed. This nonreciprocity originates in the asymmetry of vortex nucleation energy at the superconductor boundaries. Due to the sharp superconducting transition, the rectification coefficient $\Delta_{2f}$ shows a very abrupt change in the span of just 0.1 T (Fig. 2g).

In addition, this system also works as a rectenna[32,33] by picking up ambient radiation and generating d.c. voltages. The radiation pickup likely occurs in the chamber wiring of the PPMS cryostat (see Supplementary Figs. 12–15 and Supplementary Note 2). The a.c. current noise flowing through the sample wiring, and consequently through the sample, is then rectified into a d.c. voltage. The dominant frequency of the ambient radiation in our system is a few MHz (see Supplementary Fig. 4 and Supplementary Note 2).

The physical mechanism of rectification demonstrated here can be expected to appear also in other systems with topological solitons such as skyrmions in chiral magnets[34,35]. Exploiting this effect in similar systems could potentially open new directions in energy harvesting from the electromagnetic environment in novel power generation schemes or sensitive lock-in detection of electromagnetic waves.

## Methods

**Sample growth**. MoGe|YIG bilayer films were grown on 0.5-mm-thick $Gd_3Ga_5O_{12}$ (111) substrates (GGG). First, $Y_3Fe_5O_{12}$ (YIG) with a thickness of 1 μm was fabricated by liquid phase epitaxy on the GGG substrates. The YIG(111) film has an out-of-plane magnetic anisotropy[36], and its saturation field in in-plane direction is less than 0.01 T[37]. Amorphous $Mo_{1-x}Ge_x$ ($x \approx 0.22$, MoGe) films were deposited by sputtering from a tuned MoGe 70:30 target. During the sputtering, the sample holder was rotated at 300 rpm and kept at room temperature by water cooling[38–42]. Samples with MoGe thickness in the range of 340–360 nm were prepared. For control experiments, a Si wafer with a thermally oxidized $SiO_2$ surface was used instead of the YIG|GGG substrate. The plateau resistivity before the superconducting transition in 350-nm-thick MoGe films on both substrates was $\rho_N = 2.6 \times 10^{-6}$ Ω m. The onset of the superconducting transition at $B = 0$ on both substrates ($\rho = 0.95\rho_N$) appeared at $T = 7.2$ K. Resistivity dropped at $T = 6.9$ K on the YIG substrates, and at $T = 6.8$ K on the Si substrates. The GL parameter is $\kappa \approx 100$ and the magnetic penetration depth is comparable to the sample thickness[43]. Typical dimensions of the samples were $2 \times 6$ mm. The data shown in the paper were measured on four MoGe|YIG samples, one MoGe|Si sample, and one MoGe|$SiO_2$|YIG sample. The experimental results shown were also confirmed in another three MoGe|YIG samples, another three MoGe|Si samples, and a MoGe|$Y_3Al_5O_{12}$ sample.

**Measurement**. Measurements were performed in a superconducting magnet (Physical Property Measurement System, Quantum Design, Inc.—PPMS). The samples were fixed with GE7031 varnish to an oxygen-free Cu-block which was thermally sinked to a PPMS puck with H20E silver paste, or attached with screws to a PPMS functional rod. Ohmic contacts were made using Au wires attached to the MoGe surface with indium contacts. Resistivity and voltage measurements were performed at each value of the magnetic field $B$. All plots of the magnetic field dependence of voltages show $B$-odd components of voltages $V$ defined as $V_{odd}(B) = \{V(B) - V(-B)\}/2$. Note that the even component of the voltage signal, which has weak reproducibility and is attributed to accidental temperature gradients in the sample, has been excluded.

The resistivity of the samples was measured by a conventional 4-terminal method using either a resistance bridge AVS47-B (Picowatt) or a lock-in amplifier LI5640 (NF Corporation). All resistivity measurements were longitudinal, and were performed in the direction transverse to the magnetic field. Phase diagrams of superconductivity in the MoGe were constructed using the 95% rule for the upper critical field, $\rho(B_{c2}) = 0.95\rho_N$[38], and the vortex-solid melting field $B_m$ was determined as the field value where the resistivity of the sample falls below the experimental resolution, $\rho(B_m) = 10^{-3}\rho_N$. We are not aware of any previous works that investigate the phase diagram of MoGe in parallel magnetic fields.

A general expression for a voltage generated in a device as a response to current $I(t)$ is

$$V_{dev}(t) = \left[ \rho_0 I(t) + \Delta_{2f} I^2(t) + \dots \right] \ell/S, \qquad (1)$$

where $\rho_0$ is the direction-independent resistivity, $\Delta_{2f}$ represents the nonreciprocity in transport, $\ell$ is the distance between voltage terminals and $S$ is the cross-section of the conducting specimen.

When an a.c. current $I(t) = I_0 \sin(2\pi ft)$ is applied, the nonreciprocal component of the voltage to the second order of $I_0$ is

$$\frac{1}{2}\Delta_{2f}I_0^2[1 + \sin(4\pi ft - \pi/2)]\ell/S. \qquad (2)$$

To detect $\Delta_{2f}$, a higher harmonic lock-in method was used, where higher harmonic responses of the voltage are measured as

$$V_{nf}(t) = \frac{\sqrt{2}}{T} \int_{t-T}^{t} \sin(2\pi nfs + \theta) V_{dev}(s)\mathrm{d}s. \qquad (3)$$

The coefficient of the nonreciprocal response $\Delta_{2f}$ is determined from the second harmonic signal with the phase shift $\theta = -\pi/2$, that is, $V_{2f} = (\sqrt{2}/4)\Delta_{2f}I_0^2\ell/S$. The response to an a.c. current in such a nonlinear device generates a d.c. component in the voltage drop, $V_{dc} = (1/2)\Delta_{2f}I_0^2\ell/S$.

Measurements of second harmonic voltages were performed by inputting a continuous sine wave current with the frequency $f$ into the sample from a current source Keithley K6221. The amplitude and the phase of the $2f$ component of the voltage between the ends of the sample was measured by a lock-in amplifier LI5640, which was synchronized with the current source.

DC voltages were measured by connecting the two voltage terminals of the sample to a Keithley Nanovoltmeter K2181A and leaving other contacts unwired.

For control measurements with a temperature gradient, the sample was sandwiched by two sapphire plates. Temperature difference, $\Delta T$, between the two sapphire plates was generated using a chip resistance heater with negligible magnetoresistance. Temperatures of the top and bottom plates were monitored by two Cernox CX-1050 thermometers. The whole sandwich was attached to a Cu block with varnish.

## Data availability

The data that support the findings of this study are available from the corresponding author upon reasonable request.

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

## Acknowledgements

We thank Z. Qiu for fabricating the yttrium iron garnet films, S. Ito for performing selected area electron diffraction on the samples, and T. Kikkawa for experimental help. We would like to thank J. Ieda, S. Maekawa, and K. Sato for useful discussions. This research was supported by JST ERATO Spin Quantum Rectification Project (JPMJER1402), JSPS KAKENHI (Nos. 16J03699, 17H04806, 18H04215, and 18H04311), and MEXT (Innovative Area "Nano Spin Conversion Science" (No. 26103005)). J.L. is supported by the Japan Society for the Promotion of Science through a research fellowship for young scientists.

## Author contributions

J.L. conceived the experiments in discussion with Y.S., E.S., S.K., and S.O. J.L. constructed the experimental setup, performed the experiments, and analyzed the experimental data with input from Y.S., N. Kabeya, N. Kimura, K.I., S.K., S.O., and E.S. S.K. and S.O. fabricated the MoGe films. N.Y. and S.T. conducted the theoretical calculation. J.L., Y.S., N.Y., S.T., and E.S. wrote the manuscript. E.S. supervised the project. All authors discussed the results and commented on the manuscript.

## Additional information

**Competing interests:** The authors declare no competing interests.

