## [Peer Review File · Nature Communications]

Editorial Note: This manuscript has been previously reviewed at another journal that is not operating a transparent peer review scheme. This document only contains reviewer comments and rebuttal letters for versions considered at Nature Communications .

Reviewers' comments:

Reviewer #1 (Remarks to the Author):

The authors have responded adequately to my questions and the paper can now be published.

Reviewer #2 (Remarks to the Author):

I have previously reviewed this article and made numerous comments. The authors have made a nice attempt at addressing these, making changes or clarifying the issues. The authors have also made changes to address the two other referee reports in which some of the issues overlapped with mine and others did not. Overall the idea is nice and authors have clearly put considerable time and effort in this work. Although the effect is weak and probably limited to only the vortex liquid regime if one accepts that the experimental results they are seeing are correct that they are observing a vortex rectana than that is in my mind a novel device and is a distinct from that of experimental vortex ratchets systems that have been made so far. I note that I am not an experimentalists so at this point I have to trust the results and the reply to the other referees. The work could also be valuable for other variations on this systems. With that in mind I think the paper could be accepted to Nature Communications. If the editor chooses not to do this than I would also have no problem with the paper being published in Scientific Reports and I would not need to see the paper again.

Reviewer #3 (Remarks to the Author):

The authors have satisfactorily addressed most of the points raised in my original review in this revised manuscript. However, I do not feel it is appropriate to bury the added material on vortex ratchets and the work of Vodolazov and Peeters at the end of the Discussion on page 9. This should be discussed up front in the introduction on page 3 where the authors place their experiments in context with prior work.

Some of the authors' responses raise further questions about the work. For example Figure R4 (Fig. S13) in their Response appears to imply that rectified dc voltages are only generated when the noisy PPMS cryostat ground is DIRECTLY connected to the sample contact at the Hi end of the nanovoltmeter. This then raises the question as to whether the term rectenna (rectifying antenna) is appropriate here. By definition an antenna is a transducer for electromagnetic waves, which does not obviously seem to be what we are talking about here. The authors should justify the use of this term much more carefully.

Authors' response to Reviewer #1

[Comment 1-1] *The authors have responded adequately to my questions and the paper can now be published.*

[Response 1-1] We thank the Reviewer for this comment.

Authors' response to Reviewer #2

[Comment 2-1] I have previously reviewed this article and made numerous comments. The authors have made a nice attempt at addressing these, making changes or clarifying the issues. The authors have also made changes to address the two other referee reports in which some of the issues overlapped with mine and others did not. Overall the idea is nice and authors have clearly put considerable time and effort in this work. Although the effect is weak and probably limited to only the vortex liquid regime if one accepts that the experimental results they are seeing are correct that they are observing a vortex rectana than that is in my mind a novel device and is a distinct from that of experimental vortex ratchets systems that have been made so far. I note that I am not an experimentalists so at this point I have to trust the results and the reply to the other referees. The work could also be valuable for other variations on this systems. With that in mind I think the paper could be accepted to Nature Communications. If the editor chooses not to do this than I would also have no problem with the paper being published in Scientific Reports and I would not need to see the paper again.

[Response 2-1] We thank the Reviewer for this comment.

Authors' response to Reviewer #3

[Comment 3-1] *The authors have satisfactorily addressed most of the points raised in my original review in this revised manuscript. However, I do not feel it is appropriate to bury the added material on vortex ratchets and the work of Vodolazov and Peeters at the end of the Discussion on page 9. This should be discussed up front in the introduction on page 3 where the authors place their experiments in context with prior work.*

[Response 3-1] Following the comment, we have moved the corresponding paragraph from the discussion on P. 10 to the Introduction on P. 4.

In this work, we do not use vortex ratchet patterns,¹⁷⁻²⁰ but a simple boundary condition for vortex nucleation is used as proposed theoretically by Vodolazov and Peeters.²¹ Although such rectification was studied by edge roughness in superconducting strips,^{22,23} this is the first work to report power generation from environmental fluctuations.

[Comment 3-2] *Some of the authors' responses raise further questions about the work. For example Figure R4 (Fig. S13) in their Response appears to imply that rectified dc voltages are only generated when the noisy PPMS cryostat ground is DIRECTLY connected to the sample contact at the Hi end of the nanovoltmeter. This then raises the question as to whether the term rectenna (rectifying antenna) is appropriate here. By definition an antenna is a transducer for electromagnetic waves, which does not obviously seem to be what we are talking about here. The authors should justify the use of this term much more carefully.*

[Response 3-2] We thank the Reviewer for the comment. The presentation of Fig. S13 might have been a little misleading; this is the result of a control experiment which shows that ground noise can be responsible for the d.c. generation in our system. However, d.c. voltages are generated even when the cryostat ground is electrically INSULATED from the sample and wiring (but not shielded). We would like to stress that all data shown in the main text were measured in an insulated configuration (Fig. 2 in the main text). This d.c. voltage that "spontaneously" appears without connecting the sample to any inputs or noisy conductors motivated us to conduct the present study. The wiring of the cryostat chamber acts as an antenna that picks up fluctuations of electric fields from the cryostat ground through the insulation and, therefore, we believe that the use of the term "rectenna" is justified.

To avoid confusion, we have revised Fig. S13 (shown also below as Fig. R1) to distinguish between three different configurations: a) insulated (d.c. voltage in main text), b) insulated + quiet shield (almost no d.c. voltage), and c) grounded (large d.c. voltage).

Fig. R1 (Fig. S13 in Supplementary Information) Influence of coupling of the sample wiring to cryostat ground on the d.c. voltage generation in MoGe|YIG. a) A schematic illustration of the d.c. voltage measurement configuration for the data in the main text (Fig. 2) and in the remaining figures in Supplementary Information. The sample and wiring are insulated from the cryostat ground ("insulated"). b), c) Schematic illustrations of control experiments. To prove the speculation that a.c. currents in the sample appear due to coupling of wiring to the PPMS ground, a custom setup was constructed using a functional rod in which the sample is wired with an external coaxial twist-pair cable and surrounded by a shield.

In b), the shield and the sample wiring are insulated from the PPMS ground and the shield is connected to a quiet ground of the nanovoltmeter (“insulated + quiet shield”). In c), the sample wiring is directly connected to the PPMS cryostat ground shown as a red wire (“grounded”). d) Magnetic field dependence of the d.c. voltages at $T = 5$ K. The d.c. voltage from Fig. 2 in the main text is shown with red points (“insulated” configuration). When the sample and wiring are completely shielded with a quiet ground (“insulated + quiet shield” configuration, green points), the d.c. signal almost vanishes. On the other hand, when one electrode of the sample is connected to the PPMS cryostat ground (“grounded” configuration, black points), a large d.c. voltage (almost $10 \mu\text{V}$) is observed. Thus, the wiring picks up mainly electric noise from the cryostat ground, which can be responsible also for d.c. voltage generation in the insulated configuration a). It is noted that the slight shift in the peak position in c) compared to a) might be due to a temperature shift caused by the use of a custom sample probe.

To clarify the rectenna structure of our measurement system, we have newly added a schematic illustration as Fig. S15 to the Supplementary Information (see also Fig. R2 below).

Fig. R2 (Fig. S15). Schematic illustration of our rectenna device. The magnet power supply unit (PSU) is a noise source (Fig. S14), from which electric field fluctuations leak into the cryostat ground (GND). The wiring of the cryostat chamber acts as an antenna that picks up fluctuations of an electric field from the cryostat through the insulation (Fig. S13). The MoGe|YIG sample works as a rectifier and generates d.c. voltages from the a.c. noise current, which are detected at the nanovoltmeter.

REVIEWERS' COMMENTS:

Reviewer #3 (Remarks to the Author):

The authors have now satisfactorily responded to all the issues raised in my previous referee reports and I am happy to recommend that the revised manuscript be published in its present form.

Authors' response to Reviewer #3

[Comment 3-1] *The authors have now satisfactorily responded to all the issues raised in my previous referee reports and I am happy to recommend that the revised manuscript be published in its present form.*

[Response 3-1] We thank the Reviewer for their comments.